# Phosphorylation of conserved phosphoinositide binding pocket regulates sorting nexin membrane targeting

Marc Lenoir[1], Cansel Ustunel[2], Sandya Rajesh[1], Jaswant Kaur[1], Dimitri Moreau [2], Jean Gruenberg [2] & Michael Overduin [3]

Sorting nexins anchor trafficking machines to membranes by binding phospholipids. The paradigm of the superfamily is sorting nexin 3 (SNX3), which localizes to early endosomes by recognizing phosphatidylinositol 3-phosphate (PI3P) to initiate retromer-mediated segregation of cargoes to the trans-Golgi network (TGN). Here we report the solution structure of full length human SNX3, and show that PI3P recognition is accompanied by bilayer insertion of a proximal loop in its extended Phox homology (PX) domain. Phosphoinositide (PIP) binding is completely blocked by cancer-linked phosphorylation of a conserved serine beside the stereospecific PI3P pocket. This "PIP-stop" releases endosomal SNX3 to the cytosol, and reveals how protein kinases control membrane assemblies. It constitutes a widespread regulatory element found across the PX superfamily and throughout evolution including of fungi and plants. This illuminates the mechanism of a biological switch whereby structured PIP sites are phosphorylated to liberate protein machines from organelle surfaces.

[1] School of Cancer Sciences, College of Medical and Dental Sciences, University of Birmingham, Edgbaston, Birmingham B15 2TT, UK. [2] Biochemistry Department, University of Geneva, 30 quai Ernest Ansermet, 1211 Geneva 4, Switzerland. [3] Department of Biochemistry, Faculty of Medicine & Dentistry, University of Alberta, Medical Sciences Building, Edmonton, AB T6G 2H7, Canada. These authors contributed equally: Marc Lenoir, Cansel Ustunel. Correspondence and requests for materials should be addressed to M.O. (email: overduin@ualberta.ca)

All endocytosed cargos are delivered to the early endosome, where they are sorted and transported to the plasma membrane, late endosome or TGN. The retromer relies on sorting nexin proteins in order to segregate the cargo proteins destined for the TGN or the plasma membrane. The sorting nexins associated with the retromer are distinguished by their architectures. The SNX3 subtype[1] contain a PX domain but lack multimerization motifs, and may be found in vesicular, rather than tubular, carriers. The SNX1 subtype contain a Bin-Amphiphysin-Rvs (BAR) domain[2] and form dimeric complexes capable of sensing and/or remodeling membrane curvature[3]. How either type and their attached trafficking machines dynamically engage cellular membranes to control cargo trafficking remains unclear.

SNX3 is the best understood short sorting nexin that consists only of a single conserved PX domain and no flanking structural domains. Its cellular localization is principally determined by the specific interaction of its PX domain with PI3P[4], as presented at the cytosolic leaflet of the early endosome. Mutations within the conserved PI3P pocket of SNX3 abolish the membrane binding and compromise its endosomal functions[4,5]. The binding of SNX3 to endosomes is a prerequisite for association with the retromer complex[6–8] and positions the complex for cargo packaging and transit[5]. The mechanism regulating this membrane complex assembly remains an unsolved mystery that is critical for the fidelity of subcellular trafficking[9]. The recruitment of SNX3 to the cytosolic leaflet of the early endosome is a key determinant for productive retromer formation and it serves as a potential point for regulated binding and release. Visualizing how native SNX3 engages PI3P-containing membranes provides an opportunity to explain how retromer targeting and PX protein assemblies are controlled. As PI3P is constitutively present and abundant at the early endosome where a multitude of proteins recognize it, its level of production is unlikely to determine whether proteins membrane localize. Instead, we propose that another master regulatory switch must prevail, the identity of which remains unknown.

There are no structures showing sorting nexins inserted into membranes or in a regulated state. The structure of the PX domain of the distantly related yeast Grd19p protein has been reported[10], the human retromer structure has been solved[7], and coordinates of SNX3 and SNX12 PX domains coordinates have been deposited as 2YPS and 2CSK, respectively. However, key loops and terminal elements are missing in earlier structures, membrane binding data is lacking and the regulatory elements remain undefined. Moreover, no structure of any full-length sorting nexin interacting with a membrane mimic has been determined.

Here, we have characterized solution structures of native human SNX3 protein states, elucidating their interactions with each individual membrane component in order to better understand how membrane recognition is mediated and controlled. Membrane interaction by full length SNX3 is necessarily orchestrated by the specific recognition of the PI3P head group, followed by deep insertion of proximal hydrophobic elements into the bilayer. A conserved serine phosphorylation site at the rim of the PI3P binding site proved critical for switching off the interaction. Mutant forms mimicking the phosphorylated and non-phosphorylated states show that the interaction between SNX3 and early endosomal membrane is regulated in cells by Ser72 phosphorylation, which abolishes PI3P binding in vitro. This appears to constitute an all-or-none phosphorylation switch for dynamically regulating reversible endosome attachment for retromer assembly and detachment. Moreover, this highly structured regulatory element near the PIP docking site is shared with other sorting nexins that mediate membrane trafficking, suggesting a broad utility for controlling how proteins selectively recognize organelles and move cargo.

## Results

**Solution structure of human SNX3.** In contrast to SNX-BAR proteins, which form oligomers[3], SNX3 is an obligate monomer. The full length SNX3 protein remains monomeric and monodispersed in physiological solutions, as does the closely related SNX12 PX domain based on the single peaks seen in analytical ultracentrifugation (AUC) experiments. Their sedimentation coefficients of 1.81 and 1.88 s for SNX3 and SNX12-PX, respectively, are consistent with molecular masses of 15–21 kDa for SNX3 and SNX12-PX, respectively (Supplementary Fig. 1a). This compares favorably with their theoretical molecular masses of 19.9 and 17.8 kDa. Narrow peak widths in $^1$H, $^{15}$N resolved NMR spectra (Supplementary Fig. 1b and ref. [11]) are consistent with the SNX12 and SNX3 proteins being monomeric in solution, or when bound to PI3P molecules.

The structure of the 162 residue human SNX3 protein was elucidated under physiological solution conditions. The restraints used included 4984 distances derived from the volumes of crosspeaks in $^{15}$N-resolved and $^{13}$C-resolved nuclear overhauser enhancement (NOE) spectroscopy experiments, 188 backbone dihedral angles, and 88 hydrogen bond restraints (Table 1). The structure includes a classical PX fold preceded by 25 unstructured N-terminal residues and is followed by 16 residues exhibiting irregular structure based on the NMR data (Fig. 1 and Supplementary Fig. 1c). Conversely, the C-terminus is structured and interfaces with the α1, α3, and α4 helices via the Ile150, Tyr154, and Ile159 side chains, which interdigitate with the hydrophobic core of the PX domain. This juxtaposes the flexible N-terminus and structured C-terminus far from where the membrane is engaged, thus constituting a platform for tertiary interactions with the retromer alongside the 3-stranded β sheet (Fig. 1b), consistent with previous mutational studies[6]. Hence this structure of functionally intact SNX3 reveals the entire protein's structured elements and dynamic features, including those

### Table 1 Structural restraints used to calculate solution structures of SNX3 and its micelle complexes

| Experimental restraints | |
|---|---|
| *NOE distance restraints* | |
| Unambiguous | 3605 |
| Long range ($|i − j| > 5$) | 685 |
| Medium ($4 \leq |i–j| \leq 5$) | 103 |
| Short ($2 \leq |i − j| \leq 3$) | 387 |
| Sequential | 982 |
| Internal | 1448 |
| Ambiguous | 1379 |
| Hydrogen bond restraints | 44 |
| Dihedral constraints | 188 |
| $\varphi$ | 94 |
| $\psi$ | 94 |
| *Ambiguous distance restraints* | |
| Protein-micelle (PREs) (20 Å) | Asn100ε, Phe103, Gly105, Asp107, Phe110 |
| PI3P-micelle (9.71 Å) | C8 methyl group |
| Hydrogen bonds to PI3P | Arg70, Lys95, Arg118 |
| Semi-flexible residues | Val39-Gly46, Lys95-Asp111 |
| *Interactions SNX3 to micelle*[a] | |
| Hydrogen bonds (SNX3-micelle) | Arg43, Arg99, Gln100, Arg104 |
| Non-bonded | Arg43, Gly44, Lys95, Leu98-Gln100, Pro102-Asp107 |

[a]Observed in at least 50% of the structural models

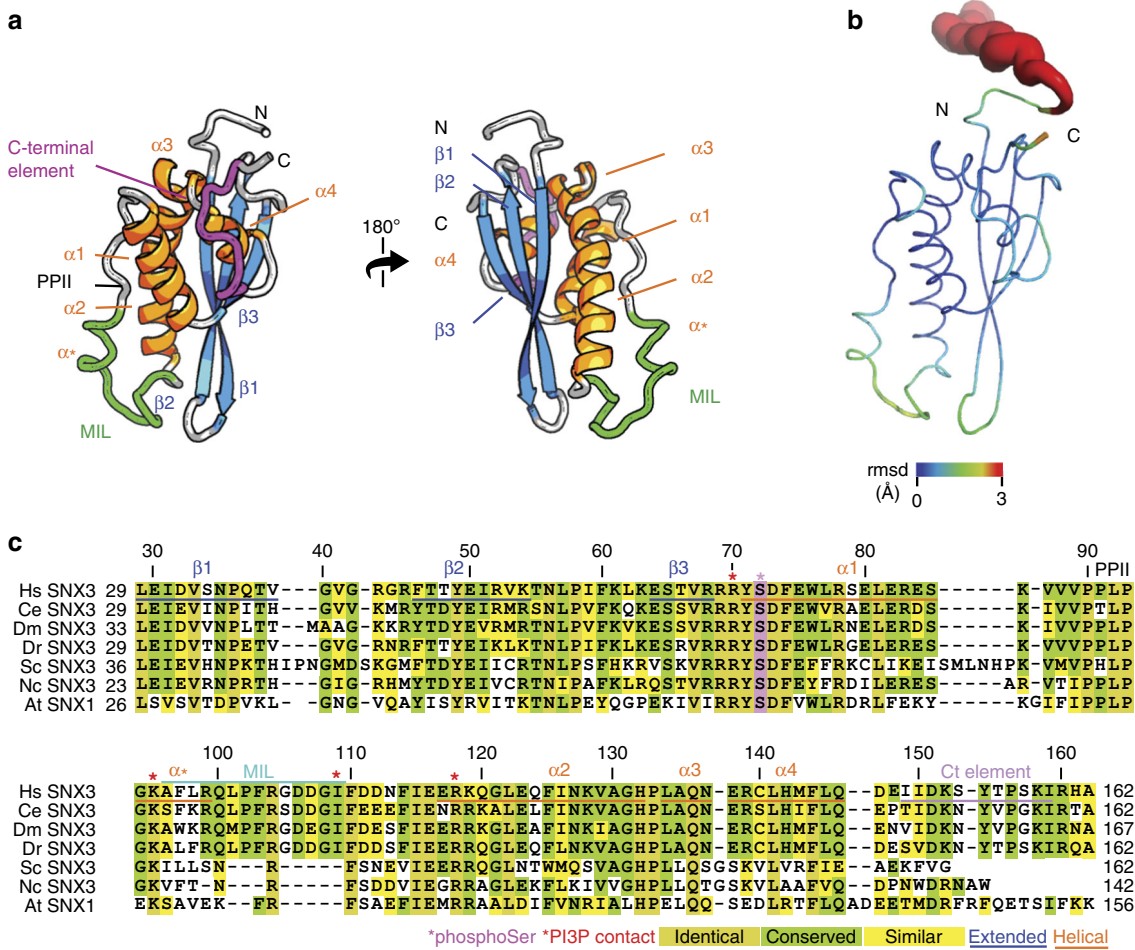

**Fig. 1** Solution structure of human SNX3. **a** The structure of SNX3 closest to the mean is shown in two perspectives with regular secondary structures, including four α-helices (orange) and three anti-parallel β-strands (blue). The segment delimited by α1 and α2-helices contains a polyproline type II helix (PPII) and α helical elements by the membrane inserting loop (MIL). The C-terminal segment is highlighted in magenta. **b** The ensemble of structures is represented as a sausage representation of the backbone traces, with the thickness and color being related to the pairwise r.m.s. deviation up to a cut-off at 3 Å (see inset scale). **c** Structure-based alignment of the SNX3 PX sequences. The C-terminally extended PX domains of SNX3 proteins from *Homo sapiens* (Hs), *Caenorhabditis elegans* (Ce), *Drosophila melanogaster* (Dm), *Danio rerio* (Dr), *Saccharomyces cerevisiae* (Sc, also known as Grd19p), and *Neurospora crassa* (Nc), as well as SNX1 from *Arabidopsis thaliana* (At), as there is no SNX3 homolog in this species, were aligned with Clustal[53]. Absolutely conserved, identical, and similar residues are shaded in brown, green, and yellow, respectively. The residues corresponding to Hs SNX3 Ser72 are magenta, and those that contact PI3P are indicated with a red asterisk. The positions of helices and extended elements are underlined and labeled

that are missing in earlier structures[10] or the retromer complex [7]. The packed C-terminal hydrophobic and aromatic positions are conserved throughout SNX3 vertebrate, invertebrate, and plant evolution (Fig. 1c), suggesting a pervasive need for the interactions of these core residues.

The SNX3 structural ensemble converges, with the PX domain complete with its C-terminal extension exhibiting a backbone root mean squared (r.m.s.) deviation of 0.37 Å (Fig. 1b, Table 2). An additional helix connects PPII to α2 and is supported by close Gly94:Phe97, Gly94:Leu98, and Ala96:Arg99 distances. This helix (α*) is missing from the earlier PX structures[10], but is functionally significant in that it positions exposed hydrophobic and basic side chains to form the membrane insertion loop (MIL) next to the β1–β2 hairpin loop. Both loops are positioned to contact the lipid bilayer based on analysis by the Membrane Optimal Docking Area (MODA) program[12], as validated by lipid titrations (see below). The canonical PI3P binding pocket is bounded by the β1-β2 sheet, α2 and MIL. It presents a positive surface with contributions from Arg70, Lys95, and Arg118 sidechains that line the PI3P pocket. The opposite side of the domain is overwhelmingly negatively charged, with acidic Glu30, Asp32,

Asp111, Asp112, Glu116, Glu117, and Glu123 residues contributing to SNX's electronegative pole (Fig. 2) that would naturally orient away from the negatively charged membrane surface, simultaneously positioning the basic pole towards the phospholipid bilayer surface.

**Nonspecific bilayer and specific PI3P interaction.** The step-wise mechanism by which SNX3 recognizes phospholipids to initiate retromer assembly on the membrane was examined by NMR titrations of individual components. To study initial non-specific sampling of the membrane, [15]N-labeled SNX3 was titrated with either dodecylphosphocholine (DPC) or diheptanoyl phosphatidylcholine (DHPC) mixed with CHAPS at a 3:1 molar ratio. Similar chemical shift perturbation (CSP) patterns were observed with either mixed micelle (Fig. 2). In both cases, the perturbations mapped exactly onto the MIL, as delimited by residues Ala96 and Asp111. Both micelles were tested as the DHPC induces larger perturbations that were easier to resolve, and contains a more biologically representative headgroup, while DPC forms smaller micelles at lower concentrations that allow more complete

**Table 2 Structural statistics for the solution structures of SNX3 and its micelle complexes**

| Structure statistics | |
|---|---|
| Residual experimental violations[a] | |
| NOE > 0.5 Å | 0 |
| NOE > 0.3 Å | 4 |
| Dihedral restraints >5 degrees | 0 |
| Energies (kcal mol$^{-1}$) | |
| $E_{noe}$ | 262.05 ± 12.13 |
| $E_{cdih}$ | 3.48 ± 0.94 |
| $E_{bond}$ | 64.20 ± 3.47 |
| $E_{improper}$ | 138.10 ± 9.25 |
| $E_{angle}$ | 278.56 ± 11.19 |
| $E_{vdw}$ | −314.5 ± 29.42 |
| $E_{dihe}$ | 918.78 ± 8.42 |
| Deviation from idealized geometry | |
| Bonds length (Å x 10$^{-3}$) | 4.76 ± 0.15 |
| Angles (°) | 0.60 ± 0.01 |
| Impropers angles (°) | 1.55 ± 0.06 |
| Atomic pairwise rmsd (Å)[b] | |
| Backbone atoms | 0.45 |
| Heavy atoms | 0.89 |
| Ramachandran statistics (%) | |
| Residues in core regions | 73.9 |
| Residues in allowed regions | 18.6 |
| Residues in generous regions | 3.1 |
| Residues in disallowed regions | 4.4 |
| Docked structure convergence (Å) | |
| Backbone atomic pairwise rmsd[c] | 1.36 ± 0.22 |
| Micelle center from the mean rmsd | 6.8 ± 3.8 |
| Intermolecular energies (kcal mol$^{-1}$) | |
| $E_{vdw}$ | −192.9 ± 15.3 |
| $E_{elec}$ | −455.5 ± 86.9 |
| $E_{restraints}$ (x10$^{-2}$) | 107.7 ± 53.1 |
| Buried surface area | 3403.9 ± 261.7 |
| PX-micelle distance/insertion angles | |
| r (micelle-protein centers) (Å) | 33.7 ± 1.8 |
| θ (deg) | 31.0 ± 5.6° |
| ψ (deg) | 81.1 ± 22.5° |

[a]20 lowest experimental energy structures
[b]For the entire PX domain, Ser26-Asp147
[c]SNX3 residues Ser26-Ala162
[d]Observed in at least 50% of the structural models

saturation of the complex. Large perturbations identified the key residues involved in the interaction as being Leu98, Gln100, Phe103-Gly105, and Ile109, which present exposed sidechains for concerted bilayer insertion. However, the affinity is weak in the absence of PI3P, with saturation not being reached even at a six-fold excess of micelle, and with an apparent binding affinity of over 2 mM for DHPC micelles (Table 3). Moreover, only a predominantly electrostatic interaction was apparent, with Gln100′s side-chain and Gly105 exhibiting paramagnetic relaxation enhancements (PREs) in presence of micelles spiked with equimolar 5-doxyl PC spin label (Fig. 3a). Together, this shows that SNX3 inserts weakly and reversibly *via* its MIL to PIP-free bilayer surfaces. The domain's dipole (Fig. 2) simultaneously reinforces bilayer binding through long range electrostatic forces, placing the termini and acidic cap far from the membrane surface and potentially available for retromer interaction.

The specific recognition of PI3P by [15]N-labeled SNX3 was monitored by stepwise addition of soluble forms of the lipid ligand. For maximum resolution, the inositol-1,3-diphosphate headgroup [Ins(1,3)P$_2$] or dibutanoyl (c$_4$) PI3P were added (Supplementary Fig. 3), allowing discrimination of the effect of the glycerol and acyl chain moieties. Both derivatives yielded similar CSPs that mapped to the canonical PIP binding pocket.

Significant perturbations were exhibited by residues Val39 in β1, Phe46-Thr47 in β2, Arg70-Ser72 in α1, Asp113, F114, Asn116, Arg118, Lys119 and Leu122 in α2, and by MIL residues Arg99-Leu101, Phe103, Gly105 and Asp107, while Ser72′s resonance underwent line broadening. Similar binding constants of 169.6 μM for Ins(1,3)P$_2$ and 158.9 μM for c$_4$-PI3P were observed, implying that SNX3 primarily recognizes the inositol headgroup of PI3P and has limited contacts with its glycerol moiety or acyl chains. A corollary is that altering the pocket responsible for specific PI3P headgroup recognition would offer the most potential for exerting regulatory control over SNX3-endosome association.

**PI3P anchors SNX3 deep into the bilayer**. The ternary complex was assembled by progressive addition of PI3P molecules and micelles to SNX3. Assignment of the complex involved titrating Ins(1,3)P$_2$ to levels approaching saturation of the protein:micelle complex. This allowed tracking of the resolved [1]H,[15]N signals (Supplementary Fig. 2a) and comparison with the slow exchange binding data obtained with c$_4$ and c$_8$-PI3P ligands[13] (Supplementary Fig. 2b). Perturbations seen in Arg70, Ser72, Leu92, Arg99, Phe103, Gly105, and Gln100ε resonances of the ternary complex mirrored those previously observed with micelles or PI3P derivatives alone. The similar patterns of perturbations induced by the individual and combined components show that the PI3P pocket and MIL interactions are local and could conceivably provide complementary effects on membrane specificity and affinity, respectively.

In order to structurally characterize the ternary complex, the exchange kinetics were tuned by varying the length of the PI3P acyl chains. The presence of zero, short or medium length chains progressively decreases the ligand exchange rates, with c$_4$-PI3P titration into micelle-saturated SNX3 exhibiting a slower off rate and higher affinity than the headgroup alone (Fig. 2, Supplementary Figs. 2b & 3). This reflects the stabilization provided by insertion of its acyl chains into the micelle in parallel with the headgroup binding to the SNX basic pocket. Ternary complex restraints were obtained from ligand titrations with DHPC and DPC-based while collecting[15]N-resolved and [13]C-resolved spectra with cryoprobe-equipped 800 and 900 MHz NMR spectrometers. The orientation of SNX3 on PI3P-containing micelles was defined using the intense [13]C signals, particularly from interfacial methyl groups. The resonances of the ligand-bound protein were assigned from c$_4$-PI3P titrations, yielding [13]C-resolved intermolecular NOE distances between Lys95 and H4 and H6 of PI3P, between Ile109δ1 and H1 and H3, and between Arg118γ and H4, thus defining the headgroup orientation (Fig. 3b). The slowly exchanging SNX3:micelle complex formed using c$_8$-PI3P allowed resolution of the bound and free state peaks, with introduction of 5-doxyl PC broadening the signals of MIL residues Phe103, Gly105 and Asp107 and Phe110 (Fig. 3a and Supplementary Fig. 2c) compared to non-paramagnetic controls. Thus these SNX3 elements insert deeply into PI3P-loaded micelles, in contrast with PC-only micelles. Hence, SNX3 relies on PI3P to anchor stably into endosomal membranes, while PI3P's absence leaves SNX3 free to dislodge after briefly sampling the bilayer. Comparison of the NOE patterns between the free and mixed micelle-bound SNX3 states indicated no major conformational changes, as corroborated by their similar circular dichroism (CD) spectra (Supplementary Fig. 1d) and local CSPs observed for SNX3′s states (Fig. 2, Supplementary Figs. 2 & 3). Together this yielded all the CSP, PRE, and NOE restraints needed to dock SNX3′s structure to the single PI3P and 54 DPC molecules within the micelle model (Table 1) in order to define how such sorting nexins structurally recognize the endosome surface.

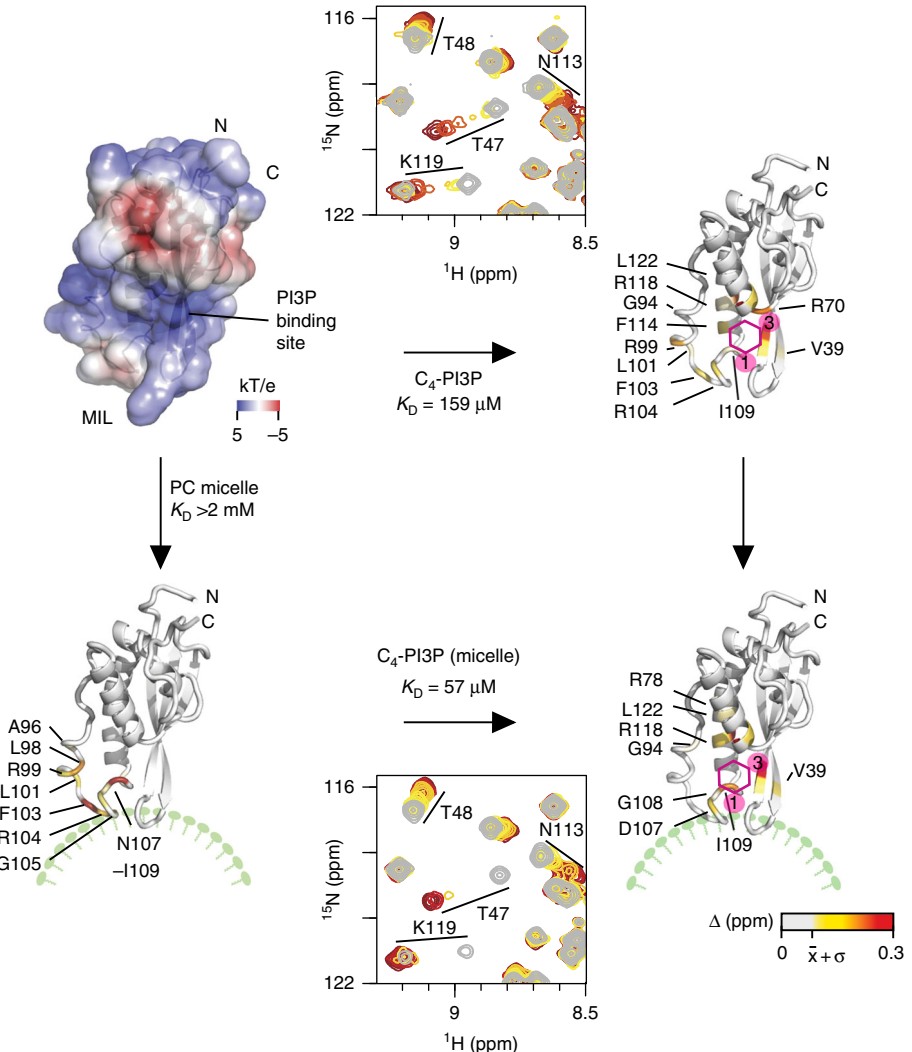

**Fig. 2** Membrane docking states of SNX3. The electrostatic surface potential of the lipid-free structure is colored blue and red for positive and negative charges, respectively, as calculated with PROPKA settings at pH 7 and APBS[54], and oriented as in Fig. 1a in the top left. The PI3P binding pocket and affinity were shown by HSQC ligand titrations, with the SNX3 ribbon colored according to the extent of $^1H$ and $^{15}N$, CSPs by addition of 4.1 fold excess $c_4$-PI3P, as drawn in magenta (top, right). The site of nonspecific micelle association is shown by the surface of the PI3P-free PX domain (below, left) colored on the basis of absolute CSPs induced by addition of 160-fold excess DPC. The position of the lipid bilayer is shown in green. The site of stable micelle association by the PI3P-bound PX domain is shown by the ribbon structure at the lower right, which is colored according to the absolute CSPs induced by addition of $c_4$-PI3P to 20-fold DPC excess. The respective affinities based on the NMR titrations are indicated, and residues whose resonances show large, medium, small and no CSPs are colored red, orange, yellow, and white, respectively

**Structural basis of membrane recognition by SNX3.** How SNX3 orients itself onto membrane surfaces to anchor retromers was calculated stepwise using NMR restraints[14]. First, the $c_8$-PI3P headgroup was docked into its binding pocket using intermolecular distances, chemical shift changes and co-crystal contacts. This showed a canonical ligand orientation, with Arg70, Lys95, and Arg118 residues forming hydrogen bonds with the inositol phosphate and hydroxyl groups, and Ile109 contacting the ring (Fig. 3b, c). The structure of SNX3 bound to $C_8$-PI3P and micelles, as calculated by HADDOCK (Fig. 3d), showed that SNX3 inserts deeply, with only 33.7 Å separating the centers of the micelle and PX domain, as opposed to the 38.7 Å separation seen in the Vam7p:micelle complex[1]. This large interface buries $3404 \pm 262$ Å$^2$ and is defined by a higher density of intermolecular

**Table 3 Affinities of SNX3 for lipid molecules and mixed micelles**

| Ligand | $K_d$ (µM)[a] |
|---|---|
| Ins(1,3)P$_2$ | 169.6 ± 34.1 |
| $c_4$-PI3P | 158.9 ± 36.1 |
| DHPC + CHAPS | >2000 |
| DH$_7$PC + CHAPS | >2000 |
| DPC + CHAPS + Ins(1,3)P$_2$ | 57.4 ± 16.4 |
| DPC + CHAPS + $c_4$-PI3P | n.d. (slow exchange) |
| DHPC + CHAPS + $c_4$-PI3P | n.d. (slow exchange) |
| DPC + CHAPS + $c_8$-PI3P | n.d. (slow exchange) |

[a]Values determined by NMR chemical shift perturbations

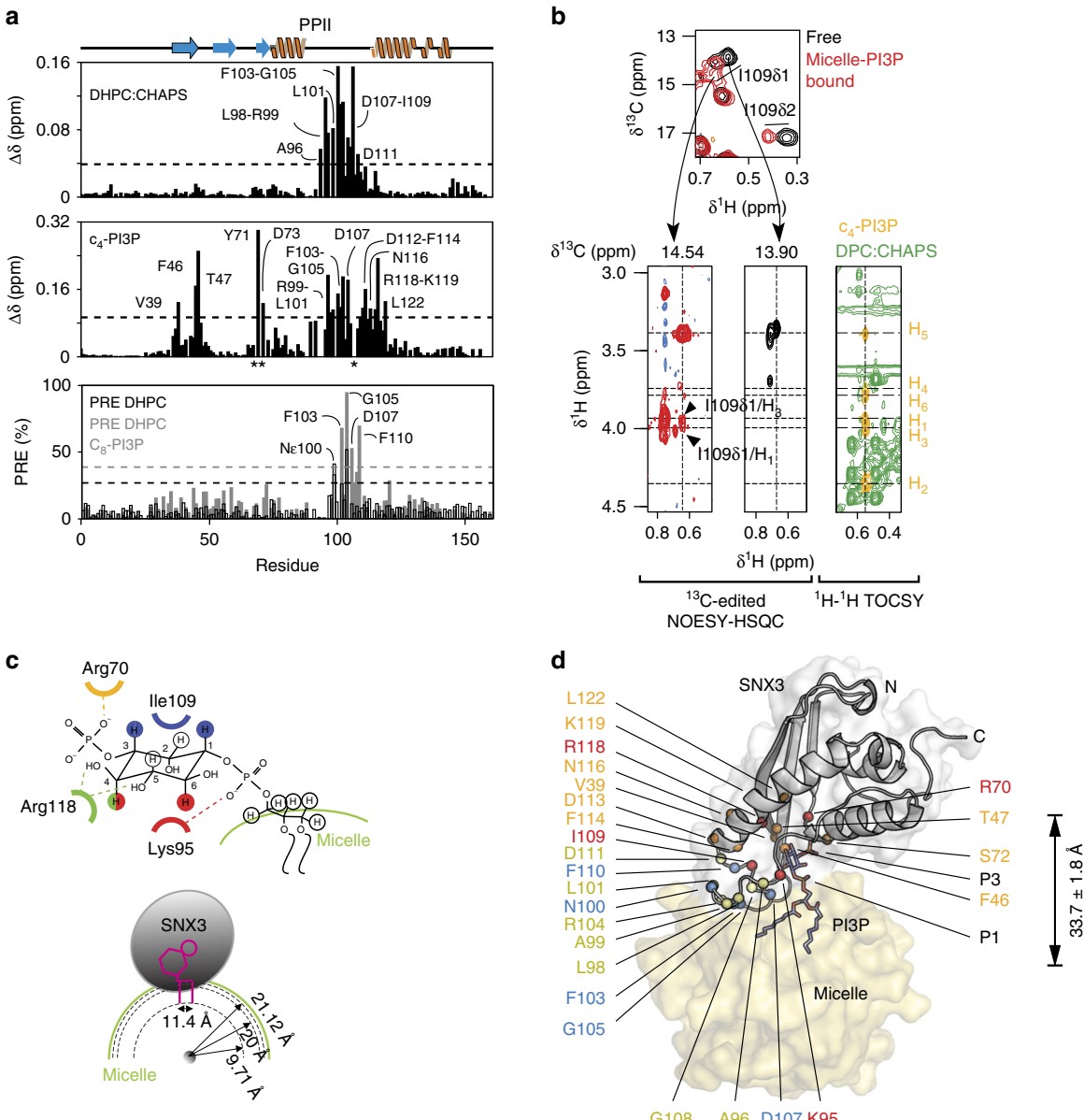

**Fig. 3** Model of SNX3 associated bound to micelles. **a** The experimentally driven molecular docking was based on the interaction of SNX3 with individual components, measuring CSPs for micelles (top) and c4-PI3P (middle). Asterisks mark residues with substantially broadened NMR lines. The membrane inserting residues were identified from PREs (bottom) obtained by adding 5 doxyl PC to 32 mM DHPC:CHAPS micelles in the absence (black) or presence of c8-PI3P (gray). **b** Strips extracted from the 1H-1H planes of 13C-edited HSQC-NOESY spectra and corresponding to Ile109's δ1 13C resonances are compared for SNX3 bound to micelles and c4-PI3P (red) or ligand-free (black). The overlaid 13C-HSQC spectra show crosspeaks of free and ligand-bound SNX3 (top). The c4-PI3P resonances (yellow) are indicated in NMR correlation spectrum alongside the DHPC: CHAPS peaks (green). These asymmetric crosspeaks correspond to intermolecular NOEs between SNX3 and PI3P. **c** The interactions between PI3P and SNX3 residues are depicted. The recognized non-exchangeable inositol protons are color-coded to match the interacting residue. Conserved hydrogen bonds observed in PX: PI3P structures are represented by dotted lines. **d** Docked structure of the lowest energy complex between SNX3, PI3P and micelles. Residues active during the docking are indicated on SNX3. Significant CSPs for Ins(1,3)P2 (orange) and/or micelles (yellow) are shown. Those involved in intermolecular interactions with PI3P (red) or micelles insertion (blue) are also displayed. The phosphate groups of PI3P are indicated by P1 and P3

restraints than has been measured for other PX domains[10,15,16] (Fig. 3d; Table 1). The SNX3 MIL is particularly long, with residues including Arg99, Gln100 or Arg104 exhibiting contacts with PI3P and seven PC molecules to orient the protein on the membrane in such a way that its β-sheet remains accessible to the retromer complex[6]. The extensive PI3P-micelle interface also includes the β1-β2 hairpin loop, with Arg43 making electrostatic contacts with PC headgroups, the specific ligand interactions of the α1 and α2 residues, and bilayer insertion of the PI3P acyl tails. Together this yields a unique insertion angle of 31.0° ± 5.6° for the

protein's long axis into the micelle interior (Table 2) using established protocols[14]. The conservation of the interfacial residues across the superfamily (Fig. 4) along with MODA-based identification of the respective membrane interaction surfaces[12] in each available PX domain structure indicates that other sorting nexins employ similar membrane binding modes.

**PI3P site phosphorylation blocks localization.** A potential mechanism of regulation of SNX3's fast and slow release from

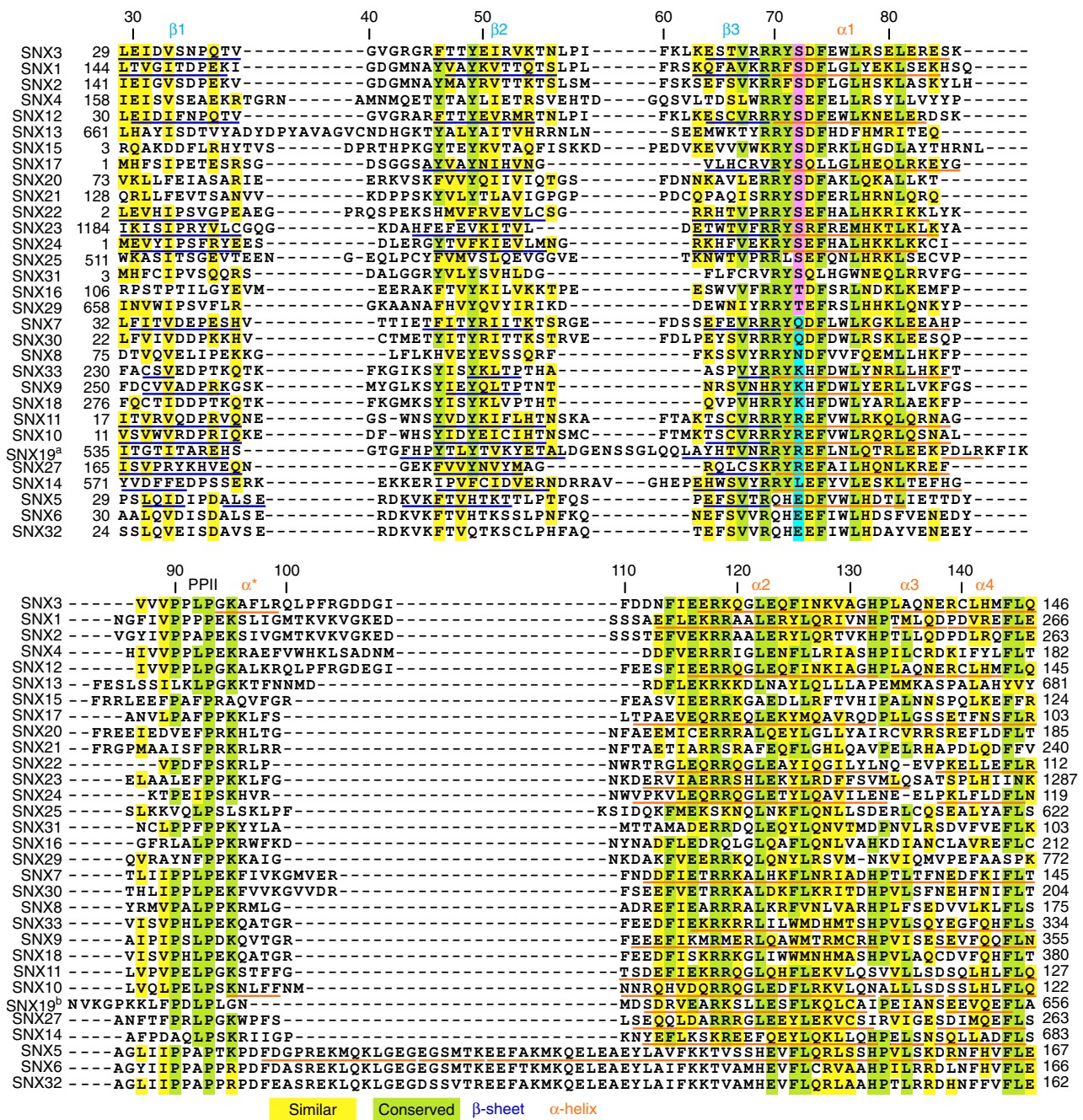

**Fig. 4** Multiple sequence alignment of SNX PX domains. Alignment of PX domain sequences from the human sorting nexin proteins. Identical, conserved and similar residues are colored green, cyan, and yellow, respectively, while the critical conserved serine residues are in magenta and substituted residues are in blue. Known secondary structure elements are indicated under the residues. All alignments were produced using the ClustalW 1.6 similarity matrix[53] and corrected to align structural elements. Numbers of the boundary amino acids are indicated on the sides, while every tenth residue of human SNX3 is indicated above

weak and strong interactions with bilayers that are either PI3P-free or loaded, respectively, was investigated. We proposed that the conserved Ser72 position (Fig. 4), which is situated near the PI3P-binding Arg70 residue (Fig. 5a & Supplementary Fig. 4), could play a crucial role. Situated at the start of α1, Ser72 is solvent-exposed, helical in structure and environmentally hypersensitive, exhibiting NMR signal variations when the pH or buffer changes. The role of Ser72 was of interest as it is predicted to be phosphorylated[17] as evidenced in various tissues[18–20], and confirmed by mass spectrometry analysis of purified SNX3

purified from HeLa cells (Supplementary Fig. 5). The SNX3 structure indicates that phosphorylation here would prevent ligand binding by blocking inositol 3-phosphate entry and negating the optimal membrane docking propensity here.

To test this PIP-stop hypothesis, SNX3 mutants mimicking constitutively non-phosphorylated (S72A) and phosphorylated (S72E) states were designed and confirmed as folded by CD (Supplementary Fig. 1d). Compared to the wild-type form, selective membrane binding activity was retained by S72A mutant but abolished by the S72E mutant (Fig. 5b, Supplementary

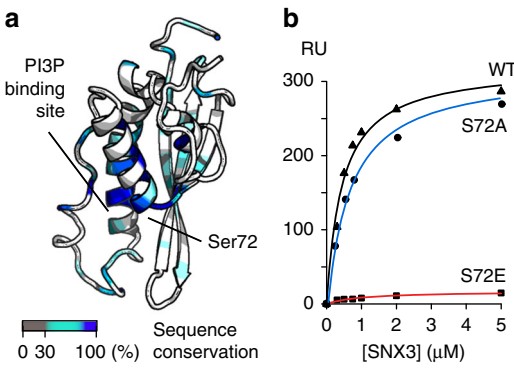

**Fig. 5** Mutation of the conserved PIP-stop residue Ser72 modulates membrane association. **a** The conservation of SNX3 residues is colored on the ribbon structure as indicated in the inset. The proximal Ser72 and PI3P pocket positions are indicated. **b** The binding affinities of wild type SNX3 and S72A and S72E mutants for PI3P-containing bilayers were measured by SPR, and show that the S72A mutation preserved activity whereas the S72E mutation abrogates binding

Figs. 6c, d). This indicates that introducing a negative charge here prevents PI3P recognition, presumably by compromising the Arg70-mediated contacts. Moreover, phosphorylation of Ser72 specifically blocks the protein from endosome docking, as validated in cellular assays. In particular, the non-interactive SNX3[S72E] mutant is cytosolic, much like the R70A mutant that cannot bind membranes because of a defective PI3P-binding site[5], whereas the constitutively membrane-active SNX3[S72A] mutant is distributed on endosomes like the wild type protein (Fig. 6). We previously showed that SNX3 overexpression inhibits endosome maturation and delays the degradation of endocytosed EGF receptor[5]. Consistent with these findings, expression of SNX3[S72A], like the WT protein, significantly delayed EGF receptor degradation, when compared to neighboring untransfected cells (Fig. 6a, c, d; Supplementary Figs. 7, 9). By contrast, the S72E mutant had essentially no effect on EGF receptor degradation, like SNX3[R70A] or free GFP (Fig. 6a, c, d). The submicromolar affinity and high specificity for PI3P-containing bilayers would retain the SNX3 protein by endosomal membranes, with its membrane-embedded Ser72 and slow off-rate leaving limited room for release. Once freed, if only transiently, protein kinases could act on the Ser72 motif in α1 to prevent SNX3 re-attachment to the membrane. Based on MODA analysis this negation of the dominant site of membrane docking in the retromer structure[7,12], this could dislodges the entire assembly from the endosome, thus allowing recycling to the TGN until reversal of the cycle by a dephosphorylation event (Fig. 7).

The PIP-stop motif is highly conserved across SNX3 homologs in plants, fungi, invertebrates and vertebrates, as well as across much of the sorting nexin superfamily (Figs. 1c and 4). Indeed, phosphorylations of the corresponding serines have been identified in proteins including SNX1 and SNX12[21,22]. To test their biological impacts, mutations of the corresponding serine residues were investigated. As predicted, the SNX1[S188E] mutant distribution was cytosolic in contrast to the endocytic localization of the wild-type form, while the control SNX12[S73A] mutant retained the wild-type punctate pattern of wild-type SNX12 (Supplementary Figs. 8, 10). Thus, they mirrored the SNX3 pattern, consistent with parallel regulatory functions. Hence, the PIP-stop mechanism appears to constitute a common way of controlling sorting nexin function, and infers a general way to modulate membrane attachment by selective phosphorylation of structured PIP binding sites.

## Discussion

Binding of sorting nexins to endosomal membranes leads to recruitment of retromers destined to the TGN[1,6] and multivesicular body genesis[5]. To understand this mechanism and its regulation, structures of SNX3 states were solved, illuminating how weak sampling and tight anchoring of bilayer surfaces occur. Membrane recognition includes a long MIL which includes an extra helix between the PPII motif and α2 that mediates both superficial, non-specific interaction with phospholipid surfaces as well as deep hydrophobic insertion into bilayers. An adjacent pocket recognizes PI3P and acts as the principal anchor, which leads to interfacial engagement of the β3-α1 loop and the beginning of α2 to form a strong membrane attachment and unique angular positioning on the surface of endosomes.

The structural characterization of the complete SNX3 protein in free, lipid-bound, micelle-inserted and mutated states exposes recurrent functional and regulatory features. The complete membrane interface of a sorting nexin has been elucidated, showing how a unique MIL element shared with SNX1, SNX2, SNX4, and SNX12 (Fig. 4) engages bilayers broadly to mediate endosomal sorting. Of these proteins SNX12 appears most like SNX3, while in SNX1 and SNX2 the hydrophobic and cationic character of the MIL is enhanced, perhaps endowing them with unique phospholipid binding modes including potential PI(3,5)P$_2$ binding[23]. A pair of unusual sorting nexins, SNX5 and SNX6, differ in their lipid binding motifs and modes, and bind to PI(4,5)P$_2$ and PI4P, respectively[24,25].

An unusual feature seen in SNX3′s solution structure is the extended C-terminal element that juxtaposes the termini. Although C-terminal extensions pack into other SNX structures, they generally involve α-helical segments as in SNX11[26] and SNX33 (pdb 4AKV), rather than the extended segment found in SNX3 as well as SNX1, SNX12 and SNX14[27]. This forms part of the retromer docking surface[7] that allows positioning a defined distance off the membrane surface (Fig. 7).

Attachment of a phosphate to a structured PI3P site was found to compete directly with phospholipid ligand binding. This PIP-stop could control membrane binding of a sorting nexin molecule or an entire retromer. Whether this event regulates the formation, cycling or disassembly of such complexes remains to be determined. Moreover, the enzymes that create PIP-stops remain a mystery, as protein kinases are usually thought to add phosphates to induce conformational changes or create protein binding sites. Here conformational changes or other proteins are not required, the PIP-stop simply repulses negatively charged lipid bilayers. This opens new perspectives for understanding how protein association with membrane targets are controlled in cellular systems. Indeed phosphates are added to the PIP binding sites of diverse sorting nexins, revealing a widespread mechanism that is conserved across most of the SNX superfamily. This post-translational modification abolishes membrane binding not only by SNX3 but also by other sorting nexins including SNX1 and SNX12, suggesting a common mechanism for controlling protein assembly and disassembly on membranes.

Our results suggest that the identity of the residue corresponding to SNX3 Ser72 could play a determining role in the specific recognition of different organelle membranes. For instance, the PX domains of SNX5 and SNX6 have glutamates at this position and are reported to recognize PI4P and PI(4,5)P$_2$ lipids[24,25], which are concentrated in the trans-Golgi network and plasma membranes, respectively. In contrast SNX14 has a leucine here, displays cytoplasmic localization[28] and lacks 3-phosphoinositide binding[27]. Thus the PIP-stop residues also could contribute to the PIP specificity and subcellular membrane destination of PX superfamily members.

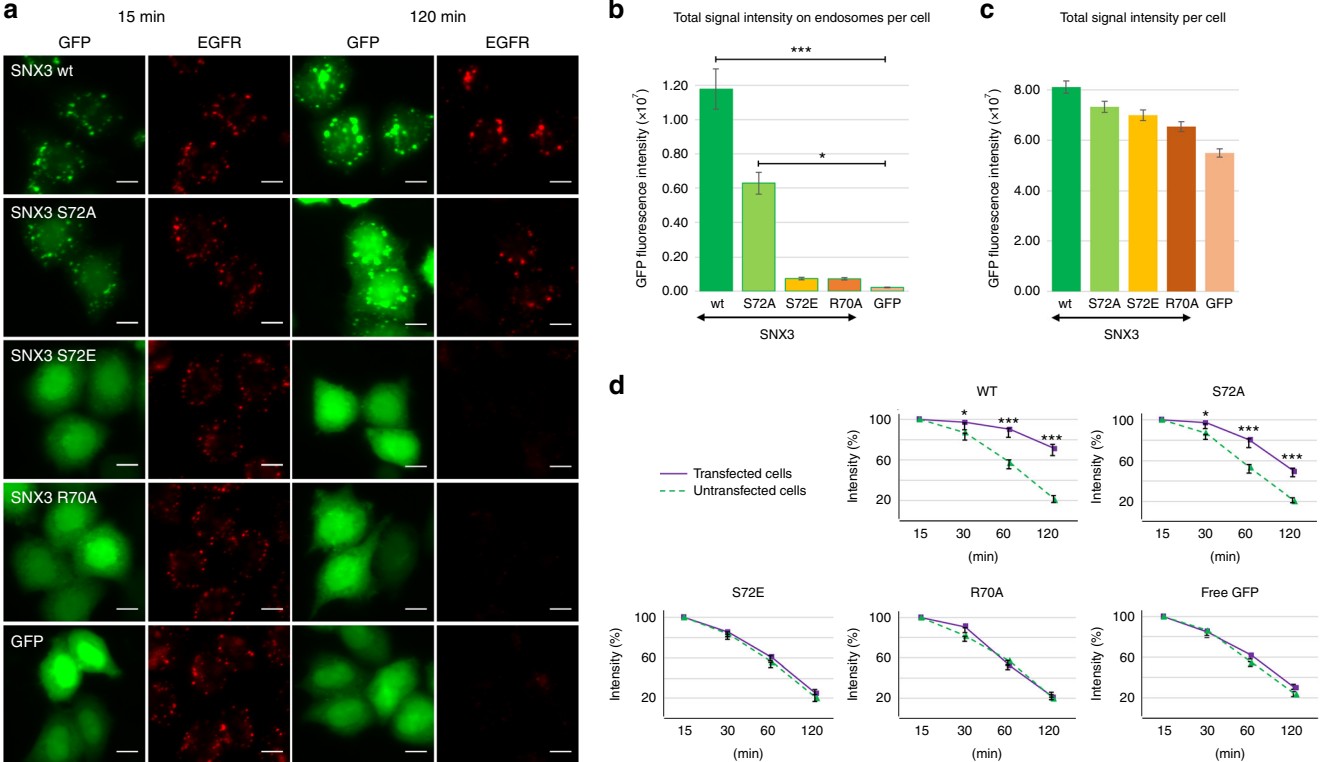

**Fig. 6** Intracellular localization of Snx3 mutants, and their effects on EGF receptor degradation. **a** Cells were transiently transfected with cDNAs coding for the indicated GFP-SNX3 fusion proteins. 18 h after transfection cells were starved for 4 h, challenged with EGF for 15 or 120 min, fixed and stained with anti-EGF receptor antibodies. The cells were then analyzed by automated microscopy. Bar: 10 μm. **b**, **c** In **a**, the total GFP fluorescence intensity of endosomes containing the EGF receptor after 15 min was quantified per cell (**b**) and is compared to the total intensity of the GFP fluorescence signal per cell (**c**). **d** The experiments was as in **a** for the indicated time periods. The total EGF receptor per cell was quantified in transfected cells (solid line) and in neighboring untransfected cells (dashed line) from the same well of the 96-well plates. The integrated intensity is expressed as a percentage of the values observed at 15 min. **b**–**d** Approximately 1000 cells were analysed per condition in $n = 3$ independent experiments. Error bars indicate SEM. Statistical significance is calculated using one-way ANOVA analysis with Bonferonni's post-test. Levels of significance are indicated as follows: $*p < 0.05$; $***p < 0.001$

Diverse species consistently show the importance of phosphorylation of SNX3 Ser72 or its equivalent phosphorylatable residue. In Drosophila SNX3 this is the primary phosphorylated site within the entire protein[29,30]. In budding yeast the homologous serine is the sole phosphorylated residue in the full length protein[30,31]. Indeed selective phosphorylation of SNX3 Ser72 (or its homologous residue) is particularly conserved throughout evolution[32]. We propose that the context of the PIP-stop, including its presence in the α1 helix by the stereospecific PIP3P pocket, predisposes this element to preferential phosphorylation over other sequences such as those that are disordered or unstructured. In other words, kinases could prefer such structured substrates, or alternatively phosphatases could find them less accessible, thus allowing PIP-stops to become common phosphorylation sites.

Cancer may preferentially involve deregulation of PIP-stop elements. In SNX3 hyperphosphorylation is seen on Ser72 in leukemia[20], lymphoma[33], breast cancer[34], and neuroblastoma cells[35], which would presumably alter the trafficking of diverse cargo. The corresponding residue is hyperphosphorylated in SNX1 in leukemia and breast[34], lung[36], and skin cancer[37]. Phosphorylation of this position is the most frequent modification of SNX12 including in breast cancer[36], gastric cancer[38], leukemia[20], lymphoma[33], and neuroblastoma[35]. Melanoma cells frequently exhibit phosphorylation of the corresponding serine of SNX12, SNX17, and SNX21[37]. Lung cancer cells exhibit hyperphosphorylation of the corresponding serine in SNX1, SNX2, SNX3, and SNX21[36,39]. Thus, PIP-stops appear to be frequently altered in a variety of cancers, suggesting a novel point of causation and possible intervention.

## Methods

**Protein expression and purification.** A sequence encoding full length human *SNX3* was synthesized (Genescript), cloned into a pET45b vector (Novagen) and overexpressed in *E. coli* BL21(DE3) cells (Novagen) as a N-terminally His₆-tagged protein. Bacterial cultures were grown in Luria-Bertani broth or M9 minimum media supplemented with ¹⁵NH₄Cl and ²H-labeled, ¹³C-labeled, or ¹³C-labeled glucose at 37 °C until an OD₆₀₀ of 0.6. After addition of 1 mM isopropyl β-D-1-thiogalactopyranoside (IPTG), the protein was expressed for 16 h at 18 °C. Cells were harvested by centrifugation (6000×*g*, 20 min) and resuspended in 20 mM Tris pH 7.5, 100 mM NaCl, 20 mM imidazole, 1 mM NaN₃ and 1 mM DTT. The cells were lysed with an Emulsiflex (Avastin) and the soluble protein was purified over a Ni²⁺-NTA affinity column (GE Healthcare). Fractions containing SNX3 were pooled and applied on a Superdex-75 (GE Healthcare) size exclusion column and eluted with 20 mM sodium phosphate buffer pH 6.5, 100 mM NaCl, 1 mM DTT. The PX domain of *SNX12* (SNX12-PX) was expressed in a pET45b vector and purified as with SNX3, except for the cleavage of the N-terminal histidine tag at an Enterokinase site prior to size exclusion chromatography. Full length *SNX1* was cloned into a pGEX-5X vector and the GST-tagged protein was purified using a GST-trap column (GE Healthcare), cleaved with Factor Xa and purified on a Superdex-200 column (GE Healthcare). DNA sequences are shown in Supplementary Table 1. Mutations were generated with QuikChange Lightening (Stratagene) and verified by DNA sequencing.

**NMR and structure calculation.** NMR spectra of ¹⁵N, ¹⁵N/¹³C, and ²H/¹⁵N/¹³C labeled protein were acquired at 298 K on 600 and 800 MHz Varian Inova spectrometers equipped with 5 mm cryogenic probes. The spectra were processed with NMRpipe[40] and analysed with CCPNMR[41]. The backbone ¹H, ¹⁵N, and ¹³C resonances were sequentially assigned using standard 3D experiments[11]. Distance restraints were derived from ¹⁵N-edited NOESY-HSQC and ¹³C-edited NOESY-HSQC experiments ($\tau_{mix} = 100$ ms) optimized for aromatic or aliphatic groups.

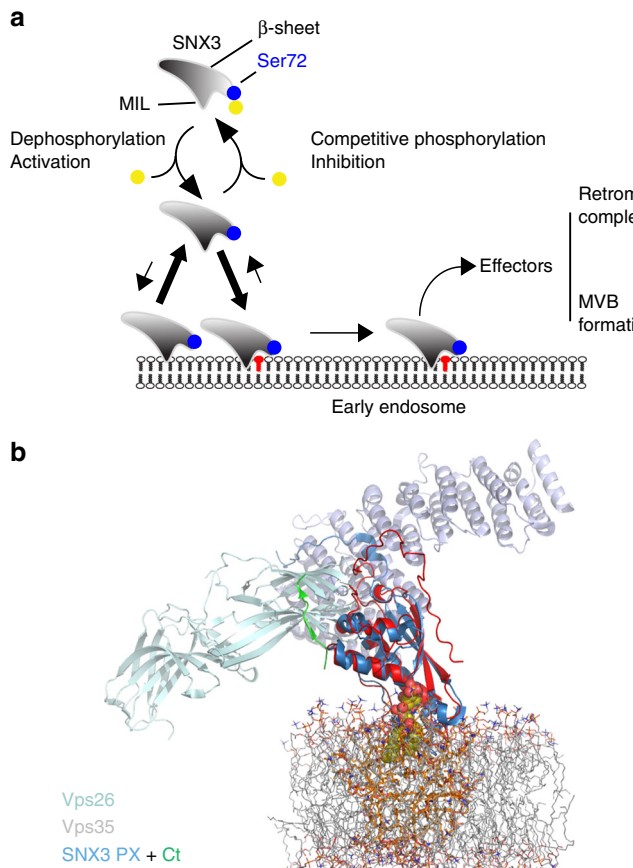

**Fig. 7** Models of regulated endosomal recruitment of sorting nexins and retromer complexes. **a** The SNX3 protein typically associates weakly with membrane, and requires PI3P to bind stably. A fraction of free SNX3 can be phosphorylated at Ser72, which prevents membrane re-binding until dephosphorylation occurs to allow tight anchoring into the endosomal membrane and retromer recruitment. **b** Model of the retromer assembly on a membrane represented by DMPC bilayer, with the solution structure of SNX3 bound to PI3P (red, with green C-terminus) superimposed on the crystal structure of SNX3 (blue) and attached Vps26 (silver-green) and Vps35 (silver-red) structures[7] depicted

Dihedral angle restraints were derived from DANGLE[42]. Hydrogen bond restraints were obtained by monitoring deuterium exchange of amide signals in HSQC spectra.

Structure calculations were performed by torsion angle dynamics in ARIA2.2 software[43]. For each of the 8 iterations, 100 structures were generated in vacuum, soaked in water and refined. The representative ensemble consisted of the 20 structures with the lowest experimental energies.

Interactions were monitored by stepwise addition of PI3P and stock solutions of DHPC or DPC micelles prepared with CHAPS in a 3:1 molar ratio. CSPs were calculated as $\Delta = [(\Delta\delta_H)^2 + (0.15 \Delta\delta_N)^2]^{1/2}$. Apparent $K_d$ values were calculated by fitting the chemical shift changes ($\Delta$) to $\Delta = \Delta_{max}(L_T + P_T + K_d - [(L_T + P_T + K_d)^2 - 4L_T P_T]^{1/2})(2P_T)^{-1}$, where $L_T$ and $P_T$ are the ligand and protein concentrations. Plots and fittings were carried out with Gnuplot.

Residues inserting into micelles were identified from PREs observed from samples containing a $c_8$-PI3P:protein ratio of 2.5:1 and DHPC:CHAPS (32:11 mM). The PRE values were calculated as $(I_{dia}-I_{para})$ $(I_{dia})^{-1}$, where $I_{dia}$ and $I_{para}$ are the SNX3 $^1$H-$^{15}$N-crosspeak intensities in the presence of micelles containing on average one diamagnetic dipalmitoyl phosphatidylcholine (DPPC) or paramagnetic 5-doxyl phosphatidylcholine (PC) molecule per micelle, respectively. Intermolecular and geometric restraints of SNX3 bound to micelles were extracted from a $^{13}$C-edited 3D NOESY experiment acquired in $D_2O$ ($\tau_{mix} = 100$ ms) from a sample containing DPC:CHAPS (8:2.7 mM) and $c_4$-PI3P (1.5 mM).

**Experimentally driven docking.** Structures of SNX3:PI3P:micelle complexes were generated according to established methods[14] using $c_8$-PI3P in the HADDOCK procedure[44]. The $c_8$-PI3P structure and parameters were designed using

XPLO2D[45] and implemented in CNSsolve[46]. The headgroup of $c_8$-PI3P was docked to SNX3 using CSP and intermolecular NOE distances as well as consistency with contacts found in SNX9 (pdb 2RAK), p40phox (pdb 1H6H), and Grd19p (pdb 1OCU) structures. Conserved hydrogen bonds between the side-chains of SNX3 and PI3P were implemented for Arg70···3-PO₄, Lys95···1-PO₄, and Arg118···4-OH/5-OH. A total of 800 SNX3 structures were calculated from which a subset of 200 were refined in explicit water. An ensemble of 20 of the lowest docking energy SNX3-PI3P structures was selected for micelle docking. Following rotational rigid body minimization of the complex, $c_8$-PI3P was embedded into the micelle. The orientation between the acyl chains of PI3P was restrained and the depth of insertion delimited according to existing DPC micelle[47] and 1-palmitoyl-2-oleoyl-PC (POPC) bilayer[48] models. Distances between the methyl's of PI3P were restrained to 11.4 Å based on dioleoyl-PC (DOPC) bilayer models to enforce realistic conformations. Distances from the center of the micelle to the acyl chain methyl were set to 9.71, and to 21.12 Å for the glycerol C3 atom, which corresponds to the average distance with the C8 and C1 atoms of DPC in micelles[47]. To handle the high temperature dynamics, the distances between guanidinium group of Arg118 and H4 and H5 atoms of PI3P were set to 3 Å. Minimizations were applied to orient the acyl chains, for rigid body insertion of the SNX3:PI3P complex into micelles and then relaxation of the system. The docking protocol yielded 200 structures which were refined in explicit water and ranked according to their energies.

**Analytical ultracentrifugation.** The SNX3 and SNX12-PX samples were prepared in the buffer used for NMR experiments and their monomeric states were evidenced using a XL-I analytical ultracentrifuge (Beckman Coulter). The sedimentation coefficient (s) and distribution [c(s)] were determined from sedimentation velocity experiments with a two-sector cell at $129,024 \times g$ for 17 h at 4 °C. The absorbance of the sample was measured at a wavelength of 280 nm throughout the cell. The partial specific volume of the protein, the viscosity and density of the buffer were calculated in SEDNTERP and used with the coefficient of sedimentation c(s) in the routine SEDFIT[49], with approximate molecular weights of the species in solution being deduced from the simplified model of continuous c (M) distribution.

**Liposome binding.** In order to assay lipid interactions POPC and PI3P (98:2 mol: mol) were mixed in chloroform and dried under a nitrogen flux and under high vacuum. Lipid mixtures (2 mM) were obtained by resuspension of the lipids into sodium phosphate 20 mM pH 7, 100 mM NaCl from which 75 μL were mixed with 25 μL of SNX3 (8 μM). After incubating the mixture for 10 min at 25 °C, the liposomes were pelleted, washed three times with the resuspension buffer and loaded on SDS-PAGE gels and bands for quantification by densitometry (Syngene). Liposomes (0.5 mM lipid) were prepared for surface plasmon resonance (SPR) detection in 50 mM Hepes pH 7.2, 50 mM NaCl. Unilamellar vesicles were produced by ten freeze and thaw cycles at 25 °C using liquid nitrogen, followed by extrusion of the vesicle through a 0.1 μm (Avanti) polycarbonate membrane. SPR measurements were carried out on a Biacore 3000 instrument (GE Healthcare). Vesicles were coated on a L1 sensor chip (GE Healthcare) at 5 μL min⁻¹. A reference lane coated with POPC alone was used to measure the specific response for lanes coated with PI3P. Equilibrium measurements were obtained by injecting 85 μL of protein at 3–5 μL min⁻¹. For kinetic measurements, the flow was increased to 30 μL min⁻¹ and 40 μL of protein were injected. Surfaces were regenerated by injections of 10 μL of 10 mM NaOH at 100 μL min⁻¹.

**Cell culture and transfection.** HeLa-MZ cells, which were provided by Marino Zerial (MPI-CBG, Dresden), and HeLa[50] cells were maintained as described. HeLa cells are not on the list of commonly misidentified cell lines maintained by the International Cell Line Authentication Committee. Our HeLa-MZ cells were authenticated by Microsynth (CH), which revealed 100 % identity to the DNA profile of the cell line HeLa (ATCC® CCL-2™) and 100 % identity over all 15 autosomal STRs to the Microsynth's reference DNA profile of HeLa. Cells are mycoplasma negative as tested by GATC Biotech (Germany). Mutations were introduced in wt SNX3-GFP[5] by QuikChange site-directed mutagenesis (Stratagene) using as primers: R70A, 5′-gaatctactgttagaagagcatacagtgactttgaatgg-3′ and 5′-ccattcaaagtcactgtatgctcttctaacagtagattc-3′; S72A, 5′-gaatctactgttagaagaga-tacgccgactttgaatggctgcgaagtgaa-3′, and 5′-ttcacttcgcagccattcaaagtcggcgtatcttcttctaa-cagtagattc-3′, S72E, 5′-gaatctactgttagaagaagatacgaggactttgaatggctgcgaagtgaa-3′ and 5′-ttcacttcgcagccattcaaagtcctcgtatcttcttctaacagtagattc-3′. Cells were transfected transiently with plasmid DNA using FuGENE HD (Promega) or TransIT-X2 Dynamic Delivery System (Mirus) according to the manufacturer's recommendations.

**Light microscopy.** HeLa-MZ transfected with wt SNX3-GFP or the indicated mutants were starved for 4 h and then incubated in the presence of EGF (100 ng mL⁻¹) for the indicated time periods. Cells were then fixed and stained with antibodies to the EGF receptor, followed by fluorescently labeled secondary antibodies. The EGF receptor was then quantified by automated microscopy using the ImageExpress microscope (Molecular Devices) after segmentation of the granular structures containing the EGFR signal[51], using Custom Module Editor™ from

MetaXpress™. The obtained mask is then applied to the EGF receptor fluorescent micrographs and the integrated intensity (sum of all pixels intensity) is extracted for each cell. To quantify the distribution of GFP-tagged SNX3 wt and mutants, the endosomes were segmented using the signal of the EGF receptor endocytosed for 15 min., The endosomes containing a significant amount of GFP signal (≥2× cytosolic background) were filtered over the entire population of structures containing the EGF receptor, in order to establish the mask for endosomes positive for both the EGF receptor and SNX3-GFP. This mask was applied to the GFP micrographs and the integrated intensity (sum of all pixels intensity) was extracted for each cell. Under all conditions, approximately 1000 cells were analysed per condition.

**Antibodies and reagents.** Mouse anti-GFP (mix of clones 7.1 and 13.1, Roche) (1:1000), rabbit anti-Vps35 (GTX108058, GeneTex) (1:1000), mouse anti-EGFR antibody (BD Transduction Laboratories) (1:200), HRP-labeled secondary antibodies (Sigma-Aldrich) (1:5000), rabbit-anti EEA1 antibody (Enzo) (1:100), fluorescently labeled secondary antibodies (Jackson ImmunoResearch Laboratories) (1:200), Hoechst (1:2500), Human EGF (Sigma). Mouse antibodies against SNX3 (C-16 Santa Cruz Biotechnology) (1:200).

**UV circular dichroism.** Proteins were dialyzed against 10 mM potassium phosphate buffer, pH 7.2. Data were acquired on a Jacso J-810 spectropolarimeter using a HELLMA cuvette with an optical path length of 0.2 mm. Signals were obtained by averaging 8 scans measured between 190 and 260 nm with a 0.2 nm increment at a sampling rate of 50 nm min$^{-1}$.

**Immunoprecipitations.** HeLa MZ cells at 60–70% confluency were transiently transfected with indicated GFP constructs; 18 h later cells were scraped in lysis buffer [30 mM Tris–HCl pH 7.4, 1 mM EDTA, 0.5% NP40 and Protease/Phosphatase Inhibitor Cocktail (CST)] and lysed. GFP-Trap MA beads (Chromotek) were used for GFP precipitation. After 45 min incubation at 4 °C with the cleared cell lysates, beads were washed with lysis buffer without detergents. Proteins attached to the beads were eluted by incubating the beads at 95 °C in 2× Laemmli buffer, and subjected to gel electrophoresis and western blotting.

**Mass spectrometry.** HeLa cells expressing SNX3-GFP were lysed and SNX3-GFP was immunoprecipitated using GFP-trap beads. The immunoprecipitate was resolved by SDS gel electrophoresis, digested with trypsin and analysed by LC-ESI-MS/MS mass spectrometry using a linear trap quadrupole Orbitrap Velos Pro (Thermo Scientific, San Jose, CA, USA) equipped with a nanoAcquity system (Waters, Milford, MA, USA). Hits were generated from raw data using the Easy-Prot software platform[52].

**Data availability.** SNX3′s coordinates have been deposited in the Protein Data Bank, with accession code 2MXC, and its NMR chemical shifts are available in BMRB Entry 25402. The authors declare that all other data supporting the findings of this study are available within the paper and its supplementary information files or from the corresponding author upon reasonable request.

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

## Acknowledgements

The authors thank Cancer Research UK, PRISM EU Sixth Framework Program, MRC, Wellcome Trust, TMIC, Genome Canada, Canada Foundation for Innovation and Campus Alberta Innovates Program for funding (M.O.), the Swiss National Science Foundation, the Swiss Sinergia program, the Polish-Swiss Research Program (PSPB-094/2010), the NCCR in Chemical Biology and LipidX from the Swiss SystemsX.ch initiative (J.G.), the HWB-NMR staff for spectrometer access and discussions, and the Wellcome Trust for support of the UK's national NMR facility.

## Author contributions

M.L. performed NMR and biochemical experiments on sorting nexins, as well as structure calculations. S.R. and J.K. produced constructs and proteins. C.U. and D.M. performed cell biology experiments. M.L., C.U., J.G., and M.O. designed experiments. M.L., C.U., J.G., and M.O. wrote the paper.

## Additional information

**Competing interests:** The authors declare no competing interests.

