## [Peer Review File · Nature Communications]

Reviewers' comments:

Reviewer #1 (Remarks to the Author):

This manuscript describes the solution structure of full-length human sorting nexin 3 (SNX3) and shows that PI3P recognition is accompanied by bilayer insertion of a proximal loop in its extended Phox homology domain. Phosphoinositide binding is blocked by cancer-linked phosphorylation of a conserved serine. This in turn releases endosomal SNX3 to the cytosol, and suggests how protein kinases might control membrane assemblies.

My overall concerns with the manuscript are that the evidence for the proposed mechanism of competitive phosphorylation is modest and the claims are overstated. The authors show that a phosphomimetic mutation interferes with PIP binding *in vitro*, and that mutations of the phosphorylation site alter the distribution of an exogenous GFP fusion in HeLa cells. However, they have no evidence that the S72E mutation is indeed phosphomimetic in this context, and the cell-based assay is described in inadequate detail to assess its rigor. For example, which promoter is the fusion expressed under? How do expression levels of the mutants compare to one another and to endogenous SNX3? Does the overexpression itself impact on endosomal sorting? Recognising that additional data in support of the mechanism may be challenging to acquire, it would be reasonable to propose the mechanism as a novel hypothesis worthy of further study. But instead the authors make much stronger claims of an unprecedented discovery that is not well supported by the data.

There are also several minor issues:

Top of page 4: "sedimentation coefficients of 1.81 and 1.88 s indicate molecular masses of 20.4 and 16.6 kDa" is inaccurate. Sedimentation coefficient and molecular mass are semi-independent results of the analysis performed here (so one does not indicate the other), and neither are determined with the claimed precision.

More generally, quantities (including errors) are quoted throughout the ms to 4 or 5 significant figures where no more than 2 or 3 are justified.

Table 2 is poorly described. The table footnote claims the values are derived from NMR lineshape analysis, yet the methods state that affinities come from titration of chemical shift perturbations, and the legend to Fig S6 suggests that SPR-derived affinities are shown in Table 2.

At face value, the data in Figure S6d suggest that the S72E mutant (not S188A mutant as labelled) binds with higher affinity than wild-type, but with ~4-fold lower R_{max} , in contradiction to the data in b and c.

Page 5: 'Moreover, only an electrostatic interaction was apparent for Gln100's sidechain and Gly105 based on paramagnetic relaxation enhancements (PRE) detection in presence of micelles spiked with equimolar 5-doxyl PC spin label (Fig. 3a).' What is the evidence that this interaction is electrostatic?

Page 6: "PI3P pocket and MIL interactions are local and provide additive effects on membrane specificity and affinity". According to Table 2, affinity is not quantified for the MIL interaction, so I can see no basis for inferring additivity here. Nor is it clear that additive effects could account for the apparent differences between DPC/CHAPS/Ins(1,3)P2 and DPC/CHAPS/c4-PI3P when there is no difference between Ins(1,3)P2 and c4-PI3P alone.

Page 7: 'Together this yields a unique insertion angle of 31° for the protein's long axis into the micelle interior.' Some comment is required on the likely accuracy and precision of the 31° angle.

Additional comments:

NMR structure calculations: how were the identities of H-bond acceptors determined?

The assigned chemical shifts should be deposited with BioMagResBank.

Figure 1b: are the rmsd mean pairwise or to the mean? This should be stated in the caption.

Page 6: 'Similar binding constants of 169.6 μM for Ins(1,3)P₂ and 158.9 μM for c4-PI3P were observed, inferring that SNX3 primarily recognizes the inositol headgroup of PI3P and has limited contacts with its glycerol moiety or acyl chains.' Inferring should be replaced with implying.

Reviewer #2 (Remarks to the Author):

Nature Communications manuscript NCOMMS-17-20614-T. Based on nmr studies of complete SNX3 protein in free, lipid- and-micelle bound forms it is concluded that the membrane association of the protein is regulated by phosphorylation of a conserved Ser residue beside de PI3P binding pocket. The novelty is that conformational changes or other proteins are not required for this regulatory switch. The loss of binding results from the fact that the negatively charged lipid bilayer is repulsed by the phosphorylation. The same theme is likely to apply for other PX superfamily proteins, and phosphorylation on this Ser residue is altered in cancer.

The manuscript is heavily centered on nmr aspects and structural work. Yet, the cell biological validation experiments on the association of SNX3 and mutants with membranes is important, well done, and convincing. The data in Figure 6 should be quantified, though.

The hyperphosphorylation of Ser72 in cancer is striking. The manuscript would gain from discussing how this regulation would be operated in normal physiological situations. Would it be part of the normal functional cycle of SNX3 (and other PX proteins)? In this case, phosphorylation and de-phosphorylation would occur during the carrier formation cycle on endosomes. Or does this phosphorylation represent a regulatory layer to tune the carrier formation frequency?

Reviewers' comments:

We thank the reviewers for their helpful comments and have strengthened the manuscript accordingly, as described below.

Reviewer #1 (Remarks to the Author):

...My overall concerns with the manuscript are that the evidence for the proposed mechanism of competitive phosphorylation is modest and the claims are overstated. The authors show that a phosphomimetic mutation interferes with PIP binding in vitro, and that mutations of the phosphorylation site alter the distribution of an exogenous GFP fusion in HeLa cells. However, they have no evidence that the S72E mutation is indeed phosphomimetic in this context, and the cell-based assay is described in inadequate detail to assess its rigor. For example, which promoter is the fusion expressed under? How do expression levels of the mutants compare to one another and to endogenous SNX3? Does the overexpression itself impact on endosomal sorting?

For clarity, the basis for the phosphomimetic mutation also includes mass spectrometry data showing that Snx3 S72 is specifically phosphorylated in cells (Figure S5 and references 18-20), and 30 years of literature showing that a glutamate substitution is the most appropriate mimic for a constitutively phosphorylated serine residue (see NA Sieracki and YA Komarova, *Studying Cell Signal Transduction with Biomimetic Point Mutations*, 2013). To further support our findings we now include significant new cellular data and methodological details.

First, each fusion protein is expressed under the control of the CMV promoter and the expression levels of each mutant is now compared to the other mutants and to the endogenous protein in new Supplementary Fig 7, and new Fig 6C. We had previously shown that overexpression of SNX3 inhibits endosome maturation and thus decreases EGF receptor degradation (reference 5, Pons et al PLoS Biol 2008), and we made use of this observation to better characterize the SNX3 mutants.

We have used automated microscopy to quantify both the distribution of the SNX3 mutants (new Fig 5A-C) and their effects on EGF receptor degradation (new Fig 5A, D) in an unbiased fashion. Cells were transfected with wt SNX3-GFP or with the mutants SNX3S72A-GFP and SNX3S72E-GFP. As controls, we also used both free GFP and SNX3R70A-GFP with a mutation in the PI3P-binding domain known to abrogate membrane binding (ref 5). Cells were challenged with EGF, and the distribution of EGF receptor was analyzed after 15min incubation at 37°C, when the receptor is predominantly found in early endosomes. New Fig 5A shows that the levels of EGFR in endosomes is comparable under all conditions, demonstrating that the overexpression of the mutants has no effect on receptor internalization. The analysis of EGFR-containing early endosomes showed that the S72A mutant is abundant on endosomes, like the wt protein, while the S72E mutant is exclusively cytosolic, like SNX3R70A that does not bind membranes and free GFP (Fig 5A, quantification in Fig 5B-C). These differences cannot be accounted for by marginal differences observed in the levels of expression of each protein (Fig 5-C).

After a longer (120min) incubation time in the presence of EGF, the EGF receptor is transported to the lysosomes and degraded. Hence, the EGF receptor signal is no longer visible in control cells expressing free GFP, SNX3S72E-GFP and SNX3R70A-GFP (Fig 5A, quantification in Fig 5D) — consistent with the findings that the S72E and R70A mutants fail to become endosome-associated (Fig 5A-B). By contrast, the EGF receptor signal is still observed in cells expressing SNX3S72A or the wt protein, presumably because the S72A mutant, like the wt protein, inhibits endosome maturation. Altogether these data further support and extend our findings by showing that the phosphomimetic S72E mutant recapitulates the effects of the cytosolic R70A SNX3 mutant, while the S72A mutant becomes endosome-associated and shares with the wt protein the capacity to inhibit endosome maturation.

Recognising that additional data in support of the mechanism may be challenging to acquire, it would be reasonable to propose the mechanism as a novel hypothesis worthy of further study. But instead the authors make much stronger claims of an unprecedented discovery that is not well supported by the data.

As requested, we have toned down our claims in the last paragraph of the introduction and in paragraph 4 of the discussion (including removal of the word “unprecedented”).

There are also several minor issues:

Top of page 4: "sedimentation coefficients of 1.81 and 1.88 s indicate molecular masses of 20.4 and 16.6 kDa" is inaccurate. Sedimentation coefficient and molecular mass are semi-independent results of the analysis performed here (so one does not indicate the other), and neither are determined with the claimed precision.

More generally, quantities (including errors) are quoted throughout the ms to 4 or 5 significant figures where no more than 2 or 3 are justified.

As suggested, we have changed the following sentence of page 4:

“Their sedimentation coefficients of 1.81 and 1.88 s indicate molecular masses of 20.4 and 16.6 kDa for SNX3 and SNX12-PX, respectively (Fig. S1a).” to:

“Their sedimentation coefficients of 1.81 and 1.88 s for SNX3 and SNX12-PX, respectively, are consistent with molecular masses of protein of 15 to 21 kDa (Fig. S1a).”

and the end of following sentence on page 13 was extended to provide clarification:

The partial specific volume of the protein, the viscosity and density of the buffer were calculated in SEDNTERP and used with the coefficient of sedimentation $c(s)$ calculated from the program SEDFIT⁴⁹, with approximate molecular weights of the species in solution being deduced from the simplified model of continuous $c(M)$ distribution.

In Figure S1a, the legend has been changed to:

a. Sedimentation velocity profiles from AUC experiments are shown for SNX3 (black) and SNX12-PX (blue), indicating monomeric states based on the protein sedimentation coefficients of 1.81 and 1.88, respectively, which are consistent with molecular masses of 15-21 kDa.

Table 2 is poorly described. The table footnote claims the values are derived from NMR lineshape analysis, yet the methods state that affinities come from titration of chemical shift perturbations, and the legend to Fig S6 suggests that SPR-derived affinities are shown in Table 2. At face value, the data in Figure S6d suggest that the S72E mutant (not S188A mutant as labelled) binds with higher affinity than wild-type, but with ~4-fold lower R_{max} , in contradiction to the data in b and c.

As suggested, we have revised the Table 2 footnote to clearly indicate that NMR chemical shift perturbations were used to estimate these affinities:

Table 2: Affinities of SNX3 for lipid molecules and mixed micelles

Ligand	$K_D(\mu M)^*$
Ins(1,3)P ₂	169.6 ±34.1
c ₄ -PI3P	158.9 ±36.1
DHPC/CHAPS	>2000

DH ₇ PC/CHAPS	>2000
DPC/CHAPS/ Ins(1,3)P ₂	57.4 ±16.4
DPC/CHAPS/c ₄ -PI3P	n.d. (slow exchange)
DHPC/CHAPS/c ₄ -PI3P	n.d. (slow exchange)
DPC/CHAPS/c ₈ -PI3P	n.d. (slow exchange)

* determined from chemical shift perturbations

Figures S6c and S6d show the SPR results which were used to estimate the affinities of SNX3 and SNX1 proteins for bilayers made of POPC/c₁₆PI3P and to compare the affinity of their mutants with those of the wild type. The respective legends have been updated to:

c. The interactions of the S72A and S72E mutant versions of human SNX3 with c₁₆-PI3P-containing bilayers were measured by SPR using a Biacore 3000 instrument. The specific response units are shown for both mutants that were injected between 0 and 5 μM protein concentrations, as represented in Figure 5.

d. The binding curves of the SNX1 S188A and S188E mutants for c₁₆-PI3P-containing bilayers based on the SPR data are shown. The phosphomimetic mutant has a dramatically reduced maximal binding level, but retains some bilayer binding presumably due to the inclusion of the BAR domain in these SNX1 constructs.

The original sensograms used to generate Fig 6d are shown below with the SNX1 S188A and S188E mutant data shown in red and grey, respectively. Both SNX1 proteins are the same molecular weight, and the phosphomimetic mutant clearly has a dramatically reduced maximal binding level. Note that fitting these curves is complex as SNX1 also contains a BAR domain, which is also able to interact with membranes and dimerize and thus has at least two lipid interaction sites. Hence, we have interpreted this data qualitatively by comparing the maximal binding level, which is consistent with the SNX3 results.

Page 5: 'Moreover, only an electrostatic interaction was apparent for Gln100's sidechain and Gly105 based on paramagnetic relaxation enhancements (PRE) detection in presence of micelles spiked with equimolar 5-doxyl PC spin label (Fig. 3a).' What is the evidence that this interaction is electrostatic?

The electrostatic nature of this interaction is based on the fact that only polar residues (and no hydrophobic residues) displayed significant paramagnetic relaxation enhancements. Given that interaction was discernibly mediated by only polar residues, we infer that the interaction is largely electrostatic with no hydrophobic residues penetrating into the hydrophobic core of the micelle. To address the concern this sentence has been changed to the following:

'Moreover, a predominantly electrostatic interaction was apparent, with Gln100's side-chain and Gly105 exhibiting paramagnetic relaxation enhancements (PRE) in presence of micelles spiked with equimolar 5-doxyl PC spin label (Fig. 3a).'

Page 6: "PI3P pocket and MIL interactions are local and provide additive effects on membrane specificity and affinity". According to Table 2, affinity is not quantified for the MIL interaction, so I can see no basis for inferring additivity here.

The affinities could not be measured quantitatively due to the slow to intermediate regime of the exchange on the NMR timescale. The reviewer is right pointing out that the affinity for the MIL is not quantified. This is because the affinity is too weak to be accurately determined, as hence is indicated as >2000uM in Table 2. Consequently, we have changed this sentence to the following:

'The similar patterns of perturbations induced by the individual and combined components show that the PI3P pocket and MIL interactions are local and could conceivably provide complementary effects on membrane specificity and affinity, respectively.'

Nor is it clear that additive effects could account for the apparent differences between DPC/CHAPS/Ins(1,3)P2 and DPC/CHAPS/c4-PI3P when there is no difference between Ins(1,3)P2 and c4-PI3P alone.

The two molecules, Ins(1,3)P2 and c4-PI3P, are very water soluble and as such, little difference would be expected (unless the critical micelle concentration of c4-PI3P was reached). As expected, we find only a small and insignificant improvement in the affinity in soluble c4-PI3P versus Ins(1,3)P2. Figure S2b illustrates how addition of c4-PI4P is capable of undergoing rapid exchange regime when binding alone, or in the presence of a micelle, binding with a remarkably slower off-rate.

Page 7: 'Together this yields a unique insertion angle of 31° for the protein's long axis into the micelle interior.' Some comment is required on the likely accuracy and precision of the 31° angle.

The docking procedure is designed to sample conformations and positions of the complexes, driven by the experimental data, and is based on published protocols described in reference 14. Based on the number of experimental restraints and the low energy ensemble of micelle/SNX3 complexes provided this represents an accurate description of how SNX3 is positioned with respect to the membranes. Even though the interacting regions are allowed to change during the docking, the calculations converged towards a unique population of

complexes, as regards as r and ϕ angle (see Table 1). The sentence has been updated to reflect this:

‘Together this yields a unique insertion angle of $31.0^\circ \pm 5.6^\circ$ for the protein’s long axis into the micelle interior (Table 1) using established protocols¹⁴.’

Additional comments:

NMR structure calculations: how were the identities of H-bond acceptors determined?

The identity of the hydrogen bond acceptors were inferred from the NOE data and from evaluation of the conserved structures of the PX domains.

The assigned chemical shifts should be deposited with BioMagResBank.

The Snx3 chemical shifts have been deposited in BioMagResBank entry 25402, as is now stated in the “Data availability” section.

Figure 1b: are the rmsd mean pairwise or to the mean? This should be stated in the caption.

The term ‘pairwise’ is stated in the revised Fig 1b legend.

Page 6: ‘Similar binding constants of 169.6 μ M for Ins(1,3)P2 and 158.9 μ M for c4-PI3P were observed, inferring that SNX3 primarily recognizes the inositol headgroup of PI3P and has limited contacts with its glycerol moiety or acyl chains.’ Inferring should be replaced with implying.

The term ‘inferring’ has been replaced with ‘implying’, as recommended.

Reviewer #2 (Remarks to the Author):

... the cell biological validation experiments on the association of SNX3 and mutants with membranes is important, well done, and convincing. The data in Figure 6 should be quantified, though.

As mentioned in our reply to Reviewer 1, we have used automated microscopy to quantify both the distribution of the SNX3 mutant (new Fig 5A-C) and their effects on EGF receptor degradation (new Fig 5A, D) in an unbiased fashion. Cells were transfected with wt SNX3-GFP or with the mutants SNX3S72A-GFP and SNX3S72E-GFP. As controls, we also used both free GFP and SNX3R70A-GFP with a mutation in the PI3P-binding domain known to abrogate membrane binding (ref 5). Cells were challenged with EGF, and the distribution of EGF receptor was analyzed after 15min incubation at 37°C, when the receptor is predominantly found in early endosomes. New Fig 5A shows that the levels of EGFR in endosomes is comparable under all conditions, demonstrating that the overexpression of the mutants has no effect on receptor internalization. The analysis of EGFR-containing early endosomes showed that the S72A mutant is abundant on endosomes, like the wt protein, while the S72E mutant is exclusively cytosolic, like SNX3R70A that does not bind membranes and free GFP (Fig 5A, quantification in Fig 5B-C). These differences cannot be accounted for by marginal differences observed in the levels of expression of each protein (Fig 5-C).

After a longer (120min) incubation time in the presence of EGF, the EGF receptor is transported to the lysosomes and degraded. Hence, the EGF receptor signal is no longer visible in control cells expressing free GFP, SNX3S72E-GFP and SNX3R70A-GFP (Fig 5A, quantification in Fig 5D) — consistent with the findings that the S72E and R70A mutants fail to become endosome-associated (Fig 5A-B). By contrast,

the EGF receptor signal is still observed in cells expressing SNX3S72A or the wt protein, presumably because the S72A mutant, like the wt protein, inhibits endosome maturation. Altogether these data further support and extend our findings by showing that the phosphomimetic S72E mutant recapitulates the effects of the cytosolic R70A SNX3 mutant, while the S72A mutant becomes endosome-associated and shares with the wt protein the capacity to inhibit endosome maturation.

The hyperphosphorylation of Ser72 in cancer is striking. The manuscript would gain from discussing how this regulation would be operated in normal physiological situations. Would it be part of the normal functional cycle of SNX3 (and other PX proteins)? In this case, phosphorylation and dephosphorylation would occur during the carrier formation cycle on endosomes. Or does this phosphorylation represent a regulatory layer to tune the carrier formation frequency?

These are excellent points. We have expanded the discussion to address these questions.

REVIEWERS' COMMENTS:

Reviewer #2 (Remarks to the Author):

The authors have responded in full to the comments that I had on the initial version of the manuscript.